# Local structural preferences in shaping tau amyloid polymorphism

Nikolaos Louros [1,2], Martin Wilkinson [3], Grigoria Tsaka[1,2], Meine Ramakers[1,2], Chiara Morelli [1,2], Teresa Garcia[1,2], Rodrigo Gallardo[3], Sam D'Haeyer[4,5], Vera Goossens[4,5], Dominique Audenaert[4,5], Dietmar Rudolf Thal [6,7], Ian R. Mackenzie[8], Rosa Rademakers [9,10], Neil A. Ranson [3], Sheena E. Radford [3], Frederic Rousseau [1,2] ✉ & Joost Schymkowitz [1,2] ✉

Tauopathies encompass a group of neurodegenerative disorders characterised by diverse tau amyloid fibril structures. The persistence of polymorphism across tauopathies suggests that distinct pathological conditions dictate the adopted polymorph for each disease. However, the extent to which intrinsic structural tendencies of tau amyloid cores contribute to fibril polymorphism remains uncertain. Using a combination of experimental approaches, we here identify a new amyloidogenic motif, PAM4 (Polymorphic Amyloid Motif of Repeat 4), as a significant contributor to tau polymorphism. Calculation of per-residue contributions to the stability of the fibril cores of different pathologic tau structures suggests that PAM4 plays a central role in preserving structural integrity across amyloid polymorphs. Consistent with this, cryo-EM structural analysis of fibrils formed from a synthetic PAM4 peptide shows that the sequence adopts alternative structures that closely correspond to distinct disease-associated tau strains. Furthermore, in-cell experiments revealed that PAM4 deletion hampers the cellular seeding efficiency of tau aggregates extracted from Alzheimer's disease, corticobasal degeneration, and progressive supranuclear palsy patients, underscoring PAM4's pivotal role in these tauopathies. Together, our results highlight the importance of the intrinsic structural propensity of amyloid core segments to determine the structure of tau in cells, and in propagating amyloid structures in disease.

The intracellular deposition of the protein tau as amyloid fibrils characterizes a heterogeneous group of more than 20 neurodegenerative diseases, called tauopathies, including Alzheimer's disease (AD), progressive supranuclear palsy (PSP) and corticobasal degeneration (CBD), among others[1]. Functional tau is most abundant in neuronal axons where it binds to microtubules, assists microtubule assembly, and is essential for the structural integrity of neurons and for axonal transport[2]. Under physiological conditions, tau is monomeric, soluble,

[1]Switch Laboratory, VIB Center for Brain and Disease Research, Herestraat 49, 3000 Leuven, Belgium. [2]Switch Laboratory, Department of Cellular and Molecular Medicine, KU Leuven, Herestraat 49, 3000 Leuven, Belgium. [3]Astbury Centre for Structural Molecular Biology, School of Molecular and Cellular Biology, University of Leeds, Leeds LS2 9JT, UK. [4]VIB Screening Core, Ghent, Belgium. [5]Centre for Bioassay Development and Screening (C-BIOS), Ghent University, Ghent, Belgium. [6]KU Leuven, Leuven Brain Institute, 3000 Leuven, Belgium. [7]Laboratory for Neuropathology, KU Leuven, and Department of Pathology, UZ Leuven, 3000 Leuven, Belgium. [8]Department of Pathology and Laboratory Medicine, University of British Columbia, Vancouver, BC, Canada. [9]Applied and Translational Neurogenomics, VIB Center for Molecular Neurology, VIB, Antwerp, Belgium. [10]Department of Biomedical Sciences, University of Antwerp, Antwerp, Belgium. ✉e-mail: Frederic.Rousseau@kuleuven.be; Joost.Schymkowitz@kuleuven.be

and intrinsically disordered[3]. Pathological conditions lead to hyperphosphorylation and other post-translational modifications of tau, resulting in loss of microtubule (MT) binding and the aggregation of tau into intracellular amyloid deposits[4,5]. Once formed, tau amyloid fibrils act as seeds that facilitate the assembly of additional tau monomers into amyloid[6,7], and such seeds can spread to functionally-connected neurons where they perpetuate tau aggregation[6]. The exact role of tau deposition and spreading in neuronal cell death remains unclear. It is currently suspected that tau amyloid fibrils or their precursors aberrantly interact with cellular components including lipids, nucleic acids, proteins, and other cellular components, leading to their role in functional dysregulation and cell death[8–10].

Different tauopathies originate in different brain regions and neuronal cell types, resulting in distinct spatio-temporal disease progression and neuropathological symptoms[11], suggesting that the pathological mechanisms underlying neurodegeneration may vary between tauopathies[12]. Recent high-resolution cryo-electron microscopy (cryo-EM) structures of amyloid fibrils in different tauopathies revealed that different tertiary amyloid folds are characteristic of different diseases, and these folds are conserved between patients with the same pathology[13]. This correlation between neuropathology and

polymorphism suggests that the amyloid structures that form reflect disease-specific pathological events[12,14]. Cellular conditions that modify tau polymorphism include post-translational modifications and alternative splicing[15], but it remains unclear how the tau sequence itself predisposes the protein to form different disease-related amyloid polymorphs. Such knowledge would be key to understanding how the amyloid structure responds to environmental perturbations, including upstream biochemical events that drive polymorph selection, and how polymorphism determines disease-specific mechanisms of toxicity, selective cellular vulnerability, and spatio-temporal patterns of spreading in the brain.

Tau consists of an N-terminal projection domain and an MT-binding domain. The MT-binding domain constitutes the core of tau amyloid deposits, which can extend towards the C-terminal tail in certain tauopathies, while the N-terminal domain remains unstructured forming a 'fuzzy coat' around the amyloid core fold[16]. The MT-binding domain is composed of a tau repeat domain (tauRD) with repeats R1 to R4 (Fig. 1a). Alternative splicing of these repeats results in isoforms that contain or lack R2[17]. Two amyloid-nucleating hexapeptide segments PHF6* ($^{275}$VQIINK$^{280}$) in R2 and PHF6 ($^{306}$VQIVYK$^{311}$) in R3 have been shown previously to drive amyloid assembly, as mutations in these

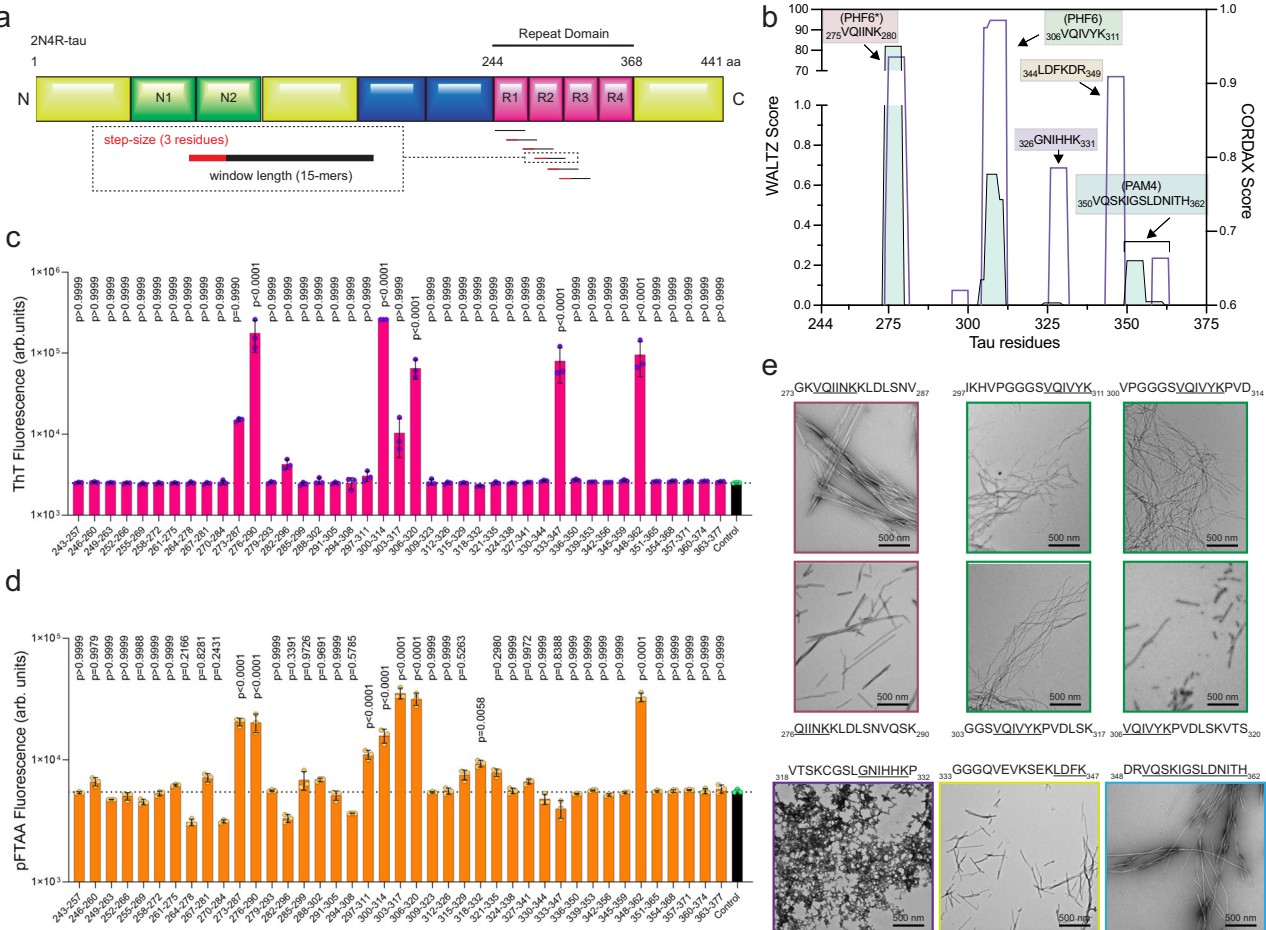

**Fig. 1 | Profiling the aggregation propensity of the tau repeat domain.**
**a** Sequence of tau (top) and sliding window approach (15-residue windows incorporating 3-residue increments) used to generate a peptide library spanning the tau repeat domain. **b** Aggregation score profile of the tau repeat domain, as predicted by CORDAX (purple line) and WALTZ (cyan area). The five regions predicted in total, including PAM4 and the previously identified PHF6* and PHF6, are highlighted in colour-shaded boxes. **c, d** End-state fluorescence analysis using Th-T (top) and pFTAA (bottom). Peptides showing increased fluorescence compared to the vehicle control (shown as a black bar) were identified as positive for

aggregation. Windows containing each one of the predicted APRs are shown in colour-shaded areas as in **b**. Bar plots indicate mean values +/− SD (*n* = 3 biologically independent samples). Statistical significance was determined using one-way ANOVA with Tukey's test for multiple comparisons. **e** Electron micrographs validating the formation of amyloid-like fibril aggregates by the identified peptide windows. Colour-coded outlines matching the shaded areas shown in **b**–**d** are used to highlight the corresponding predicted APRs contained in each peptide window. Underlined segments represent the sequences predicted in **b**. A single representative image is shown (*n* = 3 independent repeats).

sequences strongly inhibit in vitro tau amyloid formation[18,19]. However, structural analysis has shown that these segments adopt very similar conformations across tau fibril polymorphs, suggesting they do not determine polymorphism[20–23]. At the same time, no amyloid-promoting sequence has been identified to date in R4, even though this repeat is present in all disease-related human tau polymorphs.

Here, we identify a new amyloidogenic segment, PAM4 (Polymorphic AMyloid segment in repeat 4), encompassing residues 350–362 of R4, and show that it self-assembles into amyloid fibrils. Cryo-EM revealed that PAM4 fibrils are highly polymorphic, and, remarkably, that the polymorphs formed by this segment in isolation recapitulate the conformations that PAM4 adopts in polymorphs from different human tauopathies. This suggests that PAM4 possesses intrinsic structural propensities that influence the organisation of the tauRD into amyloid fibrils that favour the polymorphs observed in disease-related tau aggregates ex vivo, particularly Alzheimer's disease (AD) (3R + 4R), Corticocal Basal Degeneration (CBD) (4R) and Progressive Supranuclear Palsy (PSP) (4R) polymorphs, a notion further supported by a breakdown of residue energetic contributions in tau fibril cores. Supporting the critical importance of PAM4 in tau aggregation, PAM4-knockout cell assays show a significant impairment of tau propagation induced by seeds extracted from AD, CBD, and PSP tauopathy-associated patients. Collectively, our findings indicate that tau aggregation motifs, and PAM4 specifically, are important determinants of tau polymorphism and propagation. They also suggest that post-translational modifications, changes in splicing patterns, and the cellular environment may reinforce, rather than determine polymorphic preferences.

## Results

### Mapping amyloid propensity within the tau RD

The structure and sequence composition of the amyloid core in different tauopathies is now well-established, with all known pathologic tau deposits containing repeats R3 and R4, whereas R2 is present or absent in different disease states[14]. Amyloid sequence prediction algorithms suggest that the different tau amyloid core sequences are not uniformly aggregation-prone. The sequence-based amyloid prediction algorithm WALTZ[24] identifies two previously validated aggregation-prone regions, namely PHF6* in R2 ([275]VQIINK[280]) and PHF6 in R3 ([306]VQIVYK[311])[18,19] (Fig. 1b). However, WALTZ also predicts the existence of another amyloid motif, spanning residues [350]VQSKIGSLDNITH[362] (PAM4) in R4 of the tauRD domain, which has not been characterised previously (Fig. 1b, cyan area). To complement this analysis, we utilised CORDAX, a structure-based machine learning predictor that instead assesses the compatibility of the tauRD sequence with a cross-β structure[25]. CORDAX not only confirmed the predictions for PHF6*, PHF6, and PAM4 but also identified two additional segments, comprising [326]GNIHHK[331] and [344]LDFKDR[349] (Fig. 1b, purple line).

To determine experimentally the amyloid propensity of the entire tauRD domain, we designed a peptide library consisting of 41 peptides spanning its N- to C-terminus. This library was developed using a sliding window approach, generating 15-mer peptides with three-residue steps (Fig. 1a). For each peptide, we evaluated its amyloid-forming tendency using two distinct amyloid-reporting dyes, each with different charge:[26] the positively charged Thioflavin-T (Th-T) and the negatively charged luminescent conjugated oligothiophene (LCO) pFTAA dye. In agreement with the WALTZ and CORDAX predictions, the results from the peptide screen confirmed the binding of both dyes to specific peptides within the tauRD (Fig. 1c, d). Two neighbouring peptides, each containing PHF6* in R2 (red-shaded area) and three adjacent peptides, harbouring PHF6 in R3 (green-shaded area), exhibited binding with both dyes, with one additional peptide containing PHF6 positively binding to pFTAA but not to Th-T. Notably, the peptide spanning residues 348–362, encompassing the predicted PAM4 in R4 (cyan-shaded area) exhibited strong binding for both dyes. Electron

microscopy (EM) demonstrated that peptides containing PHF6*, PHF6, or PAM4 formed long, unbranched amyloid-like fibrils (Fig. 1e). Interestingly, neighbouring windows containing partial segments of the PAM4 motif did not exhibit any aggregation, suggesting that the intact motif is required for the formation of stable amyloid aggregates in isolation.

Two further segments were detected in the peptide screen. The first, which exhibited relatively weak binding to pFTAA, but no observable increase in ThT fluorescence, corresponds to the peptide spanning residues 318–332 (Fig. 1c, d, purple-shaded area). This region contains the [326]GNIHHK[331] sequence in R3 predicted by CORDAX. Negative stain EM showed that this peptide formed a dense network of thin amyloid-like aggregates (Fig. 1e, purple box). The second additional segment consists of the peptide spanning residues 333–347 in R4, which exclusively bound to Th-T (Fig. 1c, d, yellow-shaded area) and formed unbranched amyloid fibrils (Fig. 1e, yellow box). This peptide contains the previously predicted [337]VEVKSE[342] motif[27], as well as the [344]LDFKDR[349] sequence predicted by CORDAX (Fig. 1a). Our peptide screen thus confirms that the amyloidogenic nature of the tauRD sequence is not uniform. Instead, specific segments demonstrate an increased inherent propensity to form cross-β amyloid structure, while others clearly require additional structural context to be incorporated into the amyloid core observed for these regions in pathogenic tau fibril isolates.

### PAM4 forms amyloid fibrils that promote cellular tau seeding

To perform a deeper analysis of the amyloidogenic properties of the identified PAM4 region, we next synthesised a high purity (>99%) peptide corresponding to this segment (matching residues 350–362, Supplementary Fig. 1a, b) and characterised its ability to assemble into amyloid fibrils in more detail. The kinetics of amyloid assembly monitored using Th-T fluorescence revealed that PAM4 forms amyloid fibrils. End-state fluorescence values reveal that Th-T is bound in a concentration-dependent manner, with PAM4 showing similar Th-T levels at a matching concentration (200 μM) compared to the 348–362 peptide identified by our tau screening assay (Fig. 2a). X-ray diffraction patterns produced from oriented fibrils of the peptide were indicative of a cross-β architecture, with a meridionally oriented reflection at 4.7 Å and an equatorial reflection at 10.9 Å, corresponding to the stacking and packing distances of β-strands and β-sheets along the fibril axis, respectively (Fig. 2b). FTIR spectroscopy revealed two prominent peaks at $1631\ cm^{-1}$ and $1680\ cm^{-1}$ within the amide I region, both indicative of a β-sheet-rich conformation (Fig. 2c). Thin films containing end-state aggregates were positively stained with the Congo red dye, as seen under bright-field illumination, and exhibited green birefringence that is typical of amyloid deposits when viewed under polarised light (Fig. 2d). Atomic force microscopy (AFM) and transmission electron microscopy (TEM) revealed the formation of long, unbranched amyloid-like fibrils (Fig. 2e, f), with a higher level of polymorphism compared to the 348–362 peptide, as we could observe straight and laterally interacting fibrils, as well as twisted helical and ribbon-like morphologies (Fig. 2f).

We also investigated the amyloid nucleating properties of the same segment against the intact tauRD sequence within a cellular context. To do this, we generated seeds by sonicating mature PAM4 peptide amyloid fibrils and adding these to HEK-293 cells transiently expressing tauRD conjugated to YFP (Methods). By counting the number of cells containing yellow puncta using automated image analysis (Methods), the derived dose-response curve revealed a strong seeding efficiency of the PAM4 seeds (calculated $EC_{50} = 4.1\ \mu M$) (Fig. 2g, h). In fact, when comparing the seeding capacity of PAM4 with other amyloidogenic peptides from the tauRD identified in our screen, we found that the PAM4 peptide exhibits seeding potency 10- to 100-times higher than any of the other peptides, suggesting that it is one of the strongest tauRD nucleating regions (Fig. 2i, j).

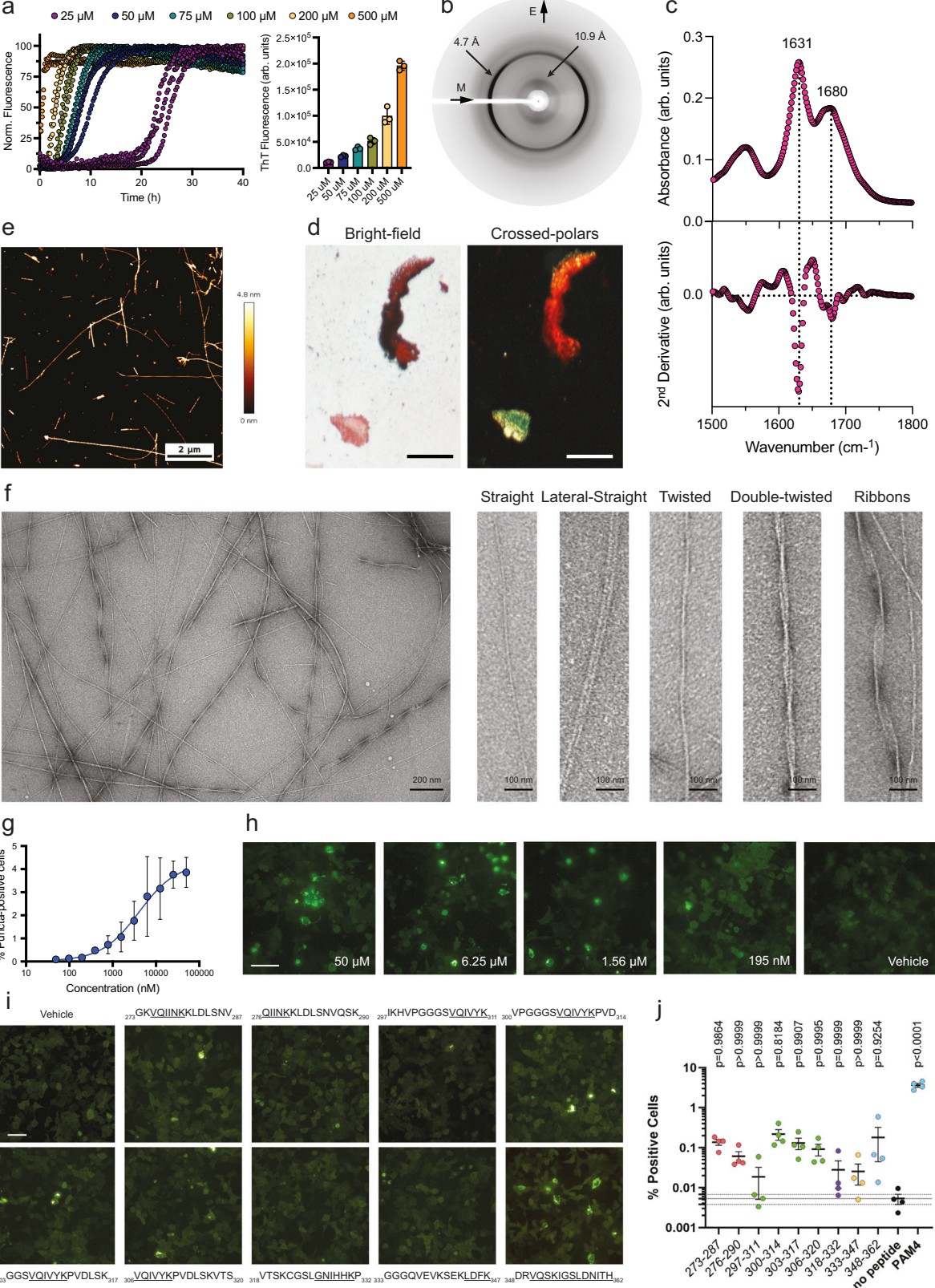

## PAM4 modulates cellular seeding of AD and other primary tauopathy-derived seeds

We next questioned the role of PAM4 in the seeding efficiency of the tauRD within a cellular context and compared its effects to the previously established PHF6* and PHF6 aggregation motifs. To do so, we transiently expressed YFP-tauRD conjugates (incorporating the P301S mutation) lacking either $^{275}$VQIINK$^{280}$ (ΔPHF6*), $^{306}$VQIVYK$^{311}$ (ΔPHF6) or $^{350}$VQSKIGSLDNITH$^{362}$ (ΔPAM4) motifs in HEK293 cells and compared their induced intracellular aggregation to that of the full-length tauRD (P301S) construct. To induce aggregation, we used seeds derived from heparin-induced recombinant tau fibrils (rTau) formed in vitro (Fig. 3a), as well as seeds formed from tau extracted from

**Fig. 2 | Biophysical characterisation of the amyloid-like properties of PAM4.**
**a** Concentration-dependent Th-T kinetic assays of the PAM4 peptide ($n = 3$ independent repeats), along with end-state Th-T fluorescence values shown in the right bar plot, with individual values shown as points. **b** Cross-β diffraction pattern produced by oriented fibres containing PAM4 peptide fibrils. Intensity quantification along the meridional and equatorial axis of the pattern indicate the presence of an intense 4.7 Å and 10.9 Å reflection, respectively. **c** FTIR spectrum produced from fibril deposits of the PAM4 peptide. The prominent 1631 cm$^{-1}$ and 1680 cm$^{-1}$ peaks are indicative of a dominant β-sheet secondary structure. **d** Polarised microscopy reveals an apple-green birefringence shown by PAM4 peptide deposits which typically signifies the presence of amyloid aggregates. (Scale Bar = 100 μM).
**e** Atomic force microscopy imaging of PAM4 fibrils. Multiple helical morphologies can be observed in a single field of view. A single representative image is shown ($n = 3$ independent repeats). **f** Electron micrograph of fibrils formed by assembly of the PAM4 peptide. Higher magnifications of individual fibrils (lower images)

showcase the presence of highly polymorphic amyloid fibrils. A single representative image is shown ($n = 3$ independent repeats). **g–h** Concentration-dependent seeding quantification performed by counting the percentage of expressing cells containing fluorescent puncta ($n = 3$ independent repeats) and representative images of the dose-dependent seeding of tauRD-YFP conjugate construct with PAM4. (Scale bar = 20 μm). **i, j** Representative images (Bar = 20 μm) and quantification of puncta-positive cells transiently expressing the tauRD-YFP construct, following transduction with 5 μM of peptide seeds. The individual points are colour-coded to match the regions highlighted in Fig. 1, with underlined segments representing the aggregation motifs predicted by Cordax and Waltz. Bar plots represent mean values ± SEM ($n = 4$ independent samples). The solid and dashed horizontal lines indicate the mean and SEM corresponding to the untreated biosensor condition. Statistical significance was determined using one-way ANOVA with Dunnett's correction for multiple comparisons.

the brains of patients diagnosed with AD (Supplementary Table 1). Dose-response curves of seeding efficiency, reported as the number of expressing cells containing fluorescent puncta, showed that tau seeding was abolished in ΔPHF6 expressing cells, regardless of the source of the seed aggregates (Fig. 3b, c, purple curves). This finding corroborates the crucial role of PHF6 in tau amyloid aggregation both in vitro and in cells[18,19]. We found that the PHF6* motif contributes to the seeding of in vitro generated tau seeds, but not for the seeds from AD patients. Specifically, a significant reduction in seeding efficiency was observed in ΔPHF6*-expressing cells when treated with rTau seeds (Fig. 3b, c, red curves), resulting in a 5-fold decrease of the EC$_{50}$ value of ΔPHF6* cells when compared to full-length tauRD (P301S) (Fig. 3d).

No change in seeding efficiency was observed when cells expressing the ΔPAM4 construct were treated with heparin-prepared seeds (Fig. 3c, d, blue triangles), suggesting that PAM4 does not contribute to this tau polymorph. However, a notable decrease in seeding efficiency was observed in these cells for seeds extracted from three independent AD patients (AD1-3) (Fig. 3b–d), where seeding potential was much higher with EC$_{50}$ values in the fM range (Fig. 3d). In all cases, deletion of PAM4 significantly impaired the ability of AD extracts to induce seeding in a cellular context, with more than a 1000-fold decrease for ΔPAM4 cells compared with cells expressing the intact tauRD (measured by their EC$_{50}$ values) (Fig. 3d). Contrastingly, seeding efficiency with AD extracts was unaffected upon deletion of PHF6*, demonstrating that this region is not important to seed AD polymorphs.

Previous structural studies have shown that R4 of tauRD (which contains PAM4) is an integral part of AD protofilaments[28], but is absent from the ordered cores of heparin-induced recombinant tau fibrils[29], while PHF6* shows the opposite organisation (present in the ordered core of heparin-induced recombinant tau fibrils). This suggests that the seeding efficacy differences observed above could potentially be explained simply by considering which segment is ordered in the different fibril structures. To investigate if this is the case, we generated cells that express tauRD in which the $^{337}$VEVKSE$^{342}$ or $^{343}$LDFKDR$^{349}$ segments are deleted, both sequences that correspond to elements identified by our aggregation screen (Fig. 1) or by previous reports[30]. Although both segments are integral parts of the structural core of AD-derived tau fibrils, these deletion constructs seeded with equal efficiency to that of the cells expressing the intact tauRD (Supplementary Fig. 2), confirming the integral role of PAM4 for templating by AD-patient tau seeds.

Next, we sought to investigate if PAM4 further modulates the seeding of polymorphs derived from other representative 3R, 4R, and 3R/4R tauopathies. Parallel cellular assays for extracts derived from a CBD, PSP, PiD and a fourth independent AD (AD4) patient case (Supplementary Table 1) once more validated the known importance of PHF6, as the ΔPHF6 deletion resulted in a ubiquitous loss of cellular

seeding regardless of the source of the tau seeds (Supplementary Fig. 3). PAM4 deletion impairs cellular seeding induced by AD (Supplementary Fig. 3a), PSP extracts (Supplementary Fig. 3b), and to a lesser extent CBD-derived extracts (Supplementary Fig. 3c). Except for PHF6, none of the other motif deletions impede cellular seeding induced by PiD-extracted aggregates compared with cells expressing the intact tauRD (Supplementary Fig. 3d). PHF6* deletion resulted in slightly increased seeding induced by PiD compared to the intact tauRD. In line with recent evidence[31], this could potentially be explained as the result of shedding segments of the fuzzy coat region, as the R2 domain is not incorporated in the core of PiD polymorphs. In conclusion, these assays demonstrate that cellular seeding propensity differs between ex vivo tau polymorphs and highlight a key role for the PAM4 segment, particularly for AD-, CBD-, and PSP-derived tau aggregates.

## Cryo-EM determination of PAM4 polymorphic protofilament folds

The structure of amyloid fibrils formed by the PAM4 peptide was explored using cryo-EM. The polymorphic propensity of PAM4, consistent with negative stain and AFM data (Fig. 2e, f), was validated by the appearance of multiple fibril morphologies in the micrographs and after 2D classification of the data (Supplementary Fig. 4a, b). Specifically, we were able to identify 2D-classes containing potential two (65%), three (28%) and four (10%) protofilament polymorphs assembled from conserved core folds (Supplementary Fig. 4c–e). The structure of the two-protofilament morphology (PAM4 Type 1) was determined at 2.6 Å resolution (Fig. 4a–c & Supplementary Fig. S5), with a helical twist of 359.32° and rise of 4.86 Å corresponding to a crossover of 125 nm (Supplementary Table 2). Each protofilament contained six PAM4 subunits, giving a total of twelve subunits per layer of the fibril with each protofilament comprising a head-to-tail ring of PAM4 peptides (Fig. 4b, c). The lower abundance, larger fibril morphologies of the PAM4 peptide fibrils could not be resolved to high resolution but appeared to contain the same protofilament building blocks as PAM4 Type 1 with increasing assembly sizes (Supplementary Figs. 4 and S5).

We also determined the structure of PAM4 amyloid fibrils formed by a lower purity (92%) peptide batch (Supplementary Fig. 1c, d). Two-dimensional (2D) classification revealed that this peptide batch also formed multiple fibril polymorphs (Supplementary Fig. 6), with three distinct PAM4 fibril structures that could be determined to high resolution (Fig. 4a–c & Supplementary Fig. S7). The most abundant polymorph, PAM4 Type 2 (representing 28% of the total population), was resolved to a resolution of 2.8 Å (Fig. 4a–c & Supplementary Fig. S7). This polymorph has six peptide subunits per layer with a helical twist and rise of 358.55° and 4.8 Å, respectively, corresponding to a crossover distance of 60 nm (Supplementary Table 2). The remaining two PAM4 fibril structures were determined at resolutions of 2.7 Å

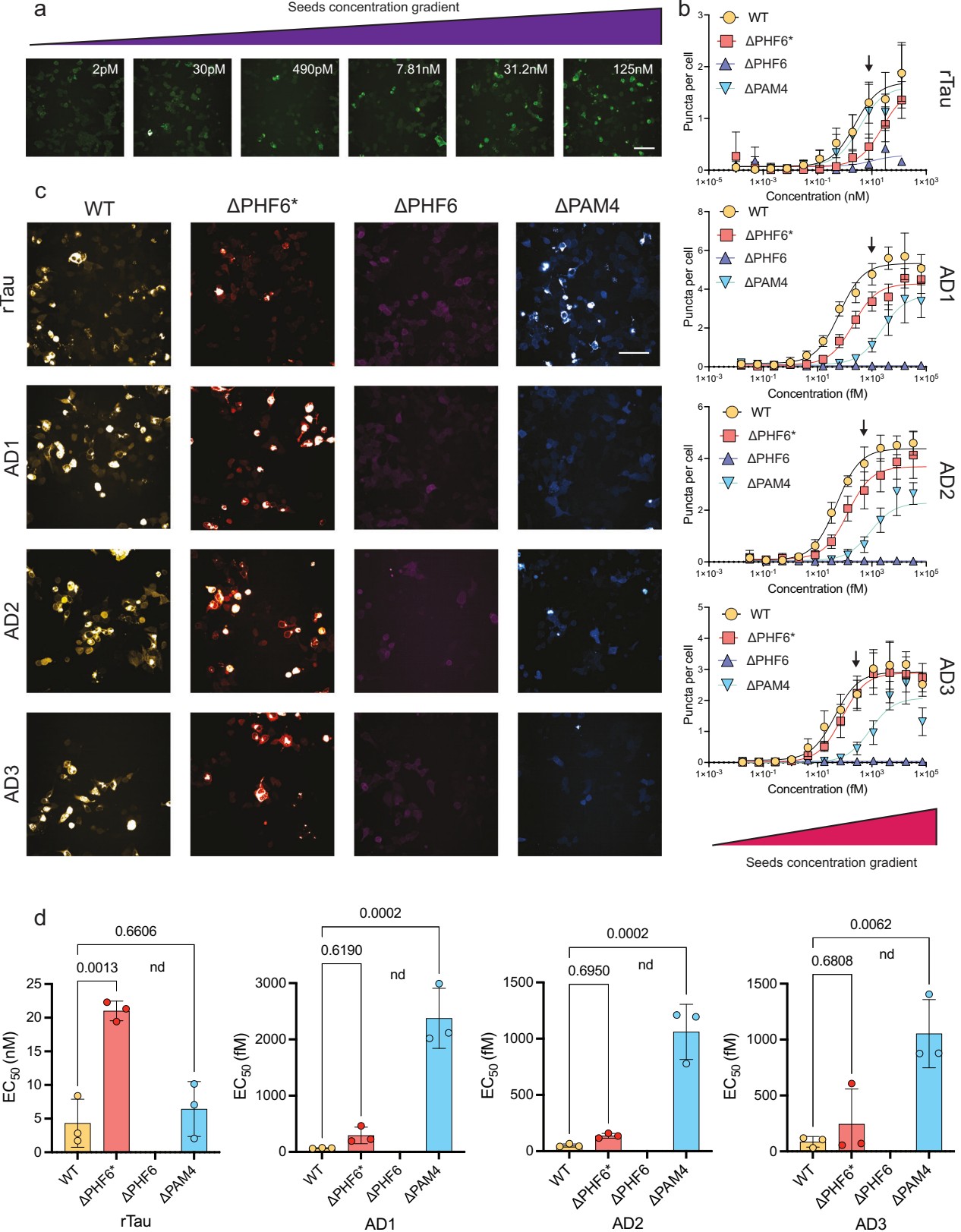

(Type 3, 15% of the total population) and 2.9 Å (Type 4, 13% of the total population), respectively (Fig. 4a–c & Supplementary Fig. S7). The Type 3 and Type 4 fibrils share similar structural properties including crossovers around 80 nm (Supplementary Table 2) and containing eight PAM4 subunits per layer of the fibril core, differing only in a slight re-positioning of the inter-protofilament interface. The

resolution of the maps, in combination with liquid chromatography-mass spectrometry (LCMS) data, facilitated the identification of an adduct coming from a minor population of the peptide containing an N-terminal Fmoc protection group from peptide synthesis (-7.4%, Supplementary Fig. 1). It is not clear what percentage of the peptide in the total population of fibrils contained this protected peptide, but in

**Fig. 3 | Cellular screening using deletion constructs of the tauRD reveals a differential relationship of APRs to tau aggregate strains. a** Intracellular seeding of cells expressing tauRD-YFP is reported by counting the number of cells with a punctuate morphology in a concentration-dependent manner upon treatment with rTau seeds. (Scale bar = 20 μm). **b** Dose-response curves after the treatment of cells expressing tauRD (WT) ΔPHF6*, ΔPHF6 or ΔPAM4 with various concentrations of rTau seeds or seeded with extracts isolated from three independent AD cases (*n* = 6, three independent samples with two technical repeats for each case). Individual points presented as mean values with ±SD. **c** Representative images of treated cells with selected seed concentrations (shown in arrow in **b**, Scale bar = 20 μm). **d** Inverted effects of the ΔPAM4 and ΔPHF6* on seeding efficiencies, shown as changes in EC$_{50}$ values, of recombinantly produced or AD extracted tau aggregates validates indicate that a bias towards specific tau polymorphs, in contrast to ΔPHF6 that is generally critical for tau aggregation. Bar plots represent mean values ± SD (*n* = 3 independent samples).

the final selected fibril segments that yielded high-resolution maps, the occupancy was high. Interestingly, when compared to N-terminal acetylation, Fmoc protection has been previously suggested to promote the formation of hydrogels by providing additional π-π stacking interactions[32]. Similarly, stacked protected N-termini pack at the protofibril interfaces of each of the three fibril structures with the C-termini extending towards the anterior fibril surface (Fig. 4c), which could potentially explain the additional structural morphologies resolved in the presence of the Fmoc protection group.

All four of the PAM4 fibril structures are inferred to be left-handed from the resolution of the cryoEM maps (Supplementary Fig. 8a), as well as from cryo-electron tomography (cryoET) reconstructions of Type 1 fibrils (Supplementary Fig. 8b, c). Altogether, PAM4 peptide monomers exhibited four unique conformations (named here Folds A-D) within the four solved fibril structures (Fig. 4d, e). Interestingly, the folds were interchangeable within the different fibril types, with Type 1 contaning FoldA and B, Type 2 fibrils containing FoldB, C and D, and Type 3 and Type 4 fibrils containing FoldC and D (Fig. 4c, d). In all cases, the folds share a C-terminal β-strand (residues 358–362) but differ in the way they orientate the N-terminus about an intermediary GS-bend (residues 355–356, Fig. 4d, e). Additionally, FoldC contains an intra-subunit salt bridge between K353 and D358 that stabilises its highly kinked (>90°) conformation (Fig. 4e). Importantly, all of the complex six-to-twelve subunit fibril layers in the PAM4 Type 1–4 polymorphs are built from repeating units of three different arrangements of the four peptide folds, which form either face-to-back or back-to-back steric zippers between their C-terminal β-strands (Fig. 4e). In the FoldA-FoldB building block (seen in PAM4 Type 1 fibrils), the FoldA subunits are similar to FoldB but bend further around the GS region to cap the end of the adjacent peptide chain (Fig. 4c, e).

## PAM4 conformations are representative of disease-associated tau polymorphs

Despite the fact that the four PAM4 amyloid peptide folds were obtained through self-assembly of synthetic peptide derivatives, superimposition onto the structures of full-length human-derived tau amyloid fibril folds revealed that the structural polymorphism of the PAM4 motif in isolation matches its conformations within the protofilaments of every major class of human tauopathies, including 3R, 4R and 3R/4R tau isoforms (Fig. 5). Specifically, the FoldC conformation of PAM4 stabilised by the K353-D358 salt-bridge is also found in tau C-fold protofilament polymorphs linked to Alzheimer's disease (AD), Gerstmann-Sträussler-Scheinker (GSS), primary age-related tauopathy (PART) and cerebral amyloid angiopathy (CAA), while showing reasonable fit to chronic traumatic encephalopathy (CTE) polymorphs (Fig. 5). FoldB PAM4, which has a flipped N-terminal compared to the previous fold, perfectly matches the 4R PSP-derived tau fibril cores and shows a C-terminal mismatch to the 3R PiD tau polymorph (Fig. 5). Finally, FoldD PAM4 matches R2:R3:R4 (4R) tau strains that are associated with corticobasal degeneration (CBD) and argyrophilic grain disease (AGD) (Fig. 5). In conjunction with our cell assays showing that PAM4 reduced AD, PSP and CBD seeding, our findings here show that the cryo-EM structures of PAM4 amyloid polymorphs in isolation are representative of the same patient-derived tau polymorphs.

## PAM4 and other motifs stabilise polymorphic tau fibril cores

We calculated and compared the per residue energy profiles of the amyloid core sequence in structurally categorised tau polymorphs (Fig. 6a–c). The cumulative per-residue energy profiles calculated for all tau fibril polymorphs indicate that specific regions play a dominant role in determining the stability of amyloid cores across different polymorphs, with other regions having relatively little stabilising influence (Fig. 6a, b). For example, PHF6* in R2, PHF6 in R3 and PAM4 in R4 consistently display stabilising effects whereas, for instance, residues 321–325 show consistent destabilization of the fibril core structure (Fig. 6b).

An examination of individual energy profiles confirms that within the context of full-length tau fibril cores and consistent with previous findings[33], PHF6 emerges as the most strong amyloid core stabilising segment across various tau structures, both in in vitro and ex vivo polymorphs (Fig. 6c). Additionally, the previously identified amyloidogenic segment PHF6* is also a key contributor to the stability of tau polymorphs in which it is present. This includes primarily in vitro-formed polymorphs and R4 polymorphs with longer protofilament cores, such as CBD/AGD and PSP/GPT/GGT cores. Although PHF6* and PHF6 are the most sSupplementary Table tructural segments in R2 and R3, respectively, this analysis also demonstrates that PAM4 emerges as the most sSupplementary Table egment of R4, and the second most stabilising region in total, in tau fibril cores polymorphs (Fig. 6b, c).

## Tau polymorphism arises from amyloid motif rearrangement into tertiary folds

The fact that the local architecture of PAM4 amyloid folds is not overruled by tertiary interactions in full-length tau amyloid cores suggests that the intrinsic structural propensity of PAM4 restricts the available conformational freedom of the tau protofilament. The way in which the PAM4 conformations are incorporated in fibril polymorphs is in good agreement with the classification recently proposed for tau[14] (Figs. 5 and 6a). R3:R4-containing polymorphs, including AD, PART and CTE protofilament folds, are stabilised by the PAM4 FoldC conformation. This L-shaped conformation of PAM4 stabilizes one leg of the horseshoe structure of these polymorphs, forming a heterotypic interface with ³²⁶GNIHHK³³¹, while PHF6 stabilizes the other leg (Fig. 6d). Interestingly, thermodynamic analysis of this PAM4 conformation in the full-length C-fold protofilaments reveals that the C-terminal segment forming this heterotypic interface with GNIHHK is the primary source of stability, with the less favourable N-terminal segment of this conformation being stabilised primarily by the formation of the K353-D358 salt-bridge, respectively (Fig. 6d).

The R1:R3:R4-containing (3R) structure of PiD also adopts a two-legged - albeit less curved - protofilament fold. In this conformation, PAM4 stabilises one leg, whereas a heterotypic interface formed between the ³³⁷VEVKSE³⁴² aggregation motif and the extremely stable PHF6 motif stabilizes the other, as also shown in previous studies[27] (Fig. 6e). The R2:R3:R4-containing (4R) tau polymorphs display a mixture of protofilament folds containing diverse conformations of the PAM4 motif. The AGD and CBD protofilament folds incorporate a longer section of the tauRD but adopt a more compact structure. These cores are stabilised centrally by a heterotypic PHF6-VEVKSE steric zipper, just as in the case of the PiD polymorph. The FoldD conformation of PAM4 stabilises the R4 segment which wraps around

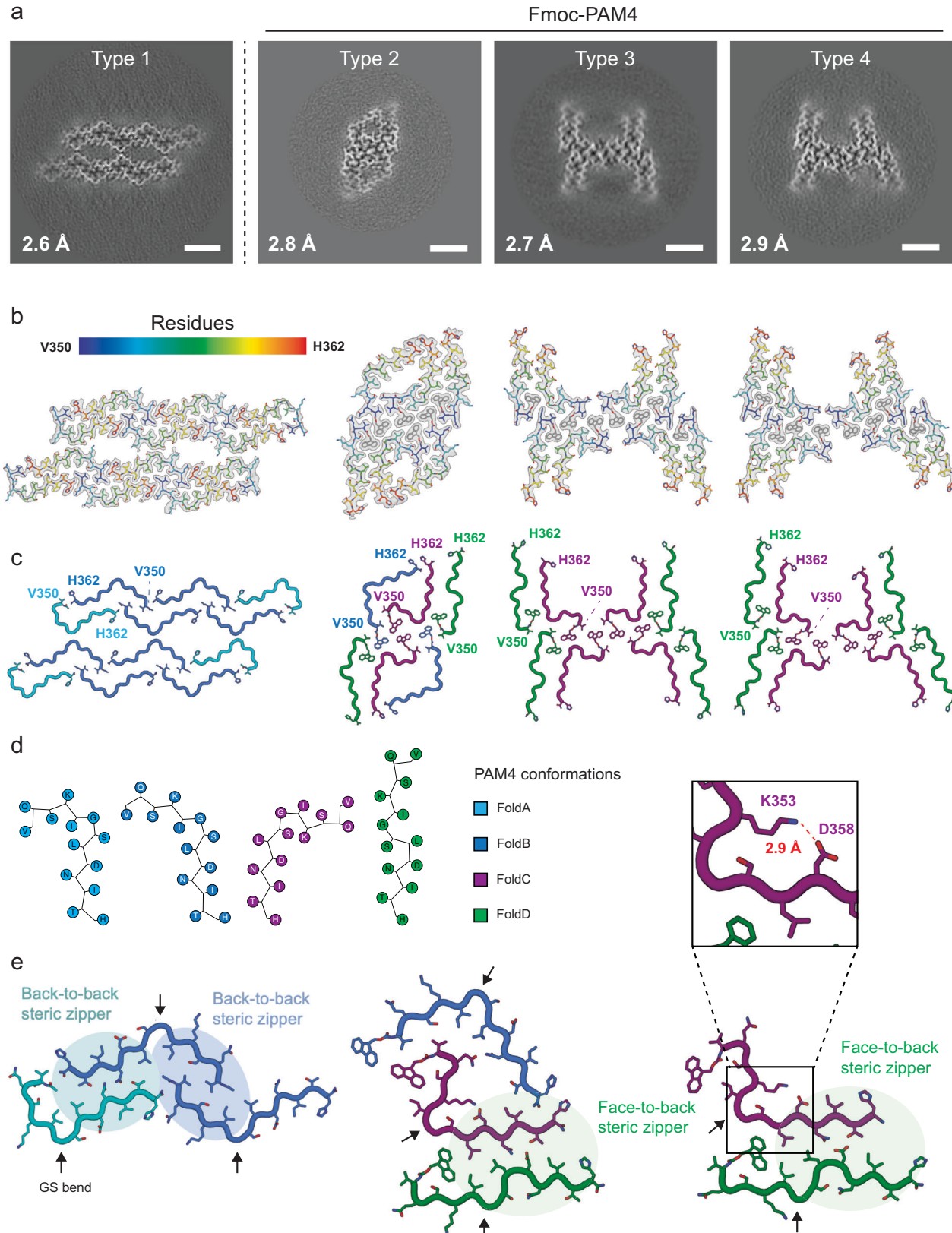

the core formed by the heterotypic zipper, with the N-terminal PHF6* closing the structure and acting as an additional molecular staple by interacting with the destabilised C-terminus of the amyloid core (Fig. 6f). In the amyloid core of fibrils derived from PSP patients, the FoldB conformation of PAM4 forms a partial heterotypic steric zipper interface of exceptional stability with PHF6 that staples one side of the

elongated fibril core, whereas the other side is stabilised through interdigitation of the PHF6*, [326]GNIHHK[331] and [337]VEVKSE[342] amyloid motifs, respectively (Fig. 6g). Finally, alternative interfaces are formed between PAM4 and PHF6 in GPT and GTT polymorphs, stabilising a particularly unstable PAM4 conformation in the latter (Fig. 6h), which may potentially explain why this PAM4 architecture was not recovered

**Fig. 4 | Cryo-EM structures of PAM4 fibrils show a diversity of folds. a** Slices through the cryo-EM map of each PAM4 fibril structure, made by averaging the central 6x slices of the post-processed, sharpened map to display approximately a single helical layer. Scale bar = 3 nm. **b** Cryo-EM maps (grey, transparent surface) with fitted atomic models for each solved structure, displayed in the same order as in **a**. Each peptide chain is coloured blue-to-red from N- to C-terminus and a single helical layer is shown. **c** PAM4 structures represented as cartoon loops, coloured by subunit fold as indicated in **d** and with the N-terminal acetyl-Val or Fmoc-Val and C-terminal amide-His residues shown as sticks and labelled. Structures are

displayed in the same order as in **a**. **d** Schematic representation of individual residues for the distinct monomeric conformations adopted by PAM4 and identified by cryo-EM. **e** Close-up views of the minimal repeating unit in each structure, displayed with cartoon backbone and stick side chains coloured by subunit as in **c**. Inter-protofilament steric zipper interactions are highlighted with shaded backgrounds, with arrows indicating the location of the GS-bend formed by Gly355 and Ser356. The insert highlights the formation of an intramolecular salt bridge between Lys353 and Asp358 that further stabilises the monomeric PAM4 FoldC.

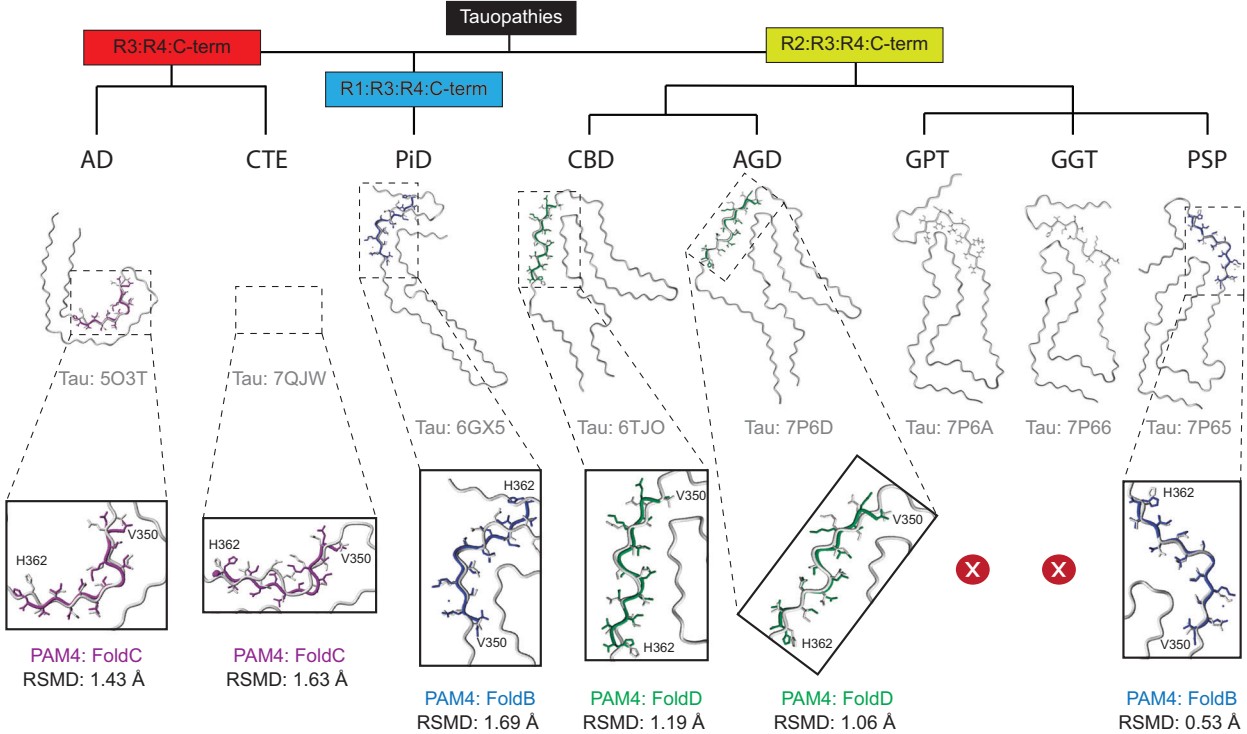

**Fig. 5 | Polymorphism of ex vivo derived tau fibrils traces back to the innate structural features of PAM4 fibrils.** The PAM4 folds recovered in isolation match its conformation in patient-derived tau amyloid fibril structures belonging to each branch of their previously proposed structural classification[14], including 3R, 4R and 3R/4R isoforms. Structural alignments reveal that the PAM4 FoldC matches the AD conformation and shows reasonable fit to CTE (3R/4R), as shown by their respective RMSD calculations (1.43 Å and 1.63 Å, respectively). Similarly, FoldB matches the

PSP conformation and shows a mismatch at the C-terminal of the region in the PiD conformation, respectively (RMSD values of 0.53 Å and 1.69 Å). Finally, FoldD perfectly overlaps the CBD and AGD (4 R) conformations (RMSD values of 1.19 Å and 1.06 Å, respectively). Zoom-in inlets highlight the superposed PAM4 segments from tau polymorphs and the individual folds observed in isolation using cryoEM. Tau structures are coloured in grey, whereas individual PAM4 folds are coloured as in Fig. 4d.

in isolation. Together, these residue energy profiles reveal that, despite their structural diversity, tau fibril cores exhibit significant similarities in terms of the structural elements governing their thermodynamic stability and that only a few specific amyloidogenic segments, including PAM4, serve as common building blocks.

## Discussion

The context in which tau isoforms are expressed at the cellular and tissue levels[21,22,29,34–36], as well as disease-specific post-translational modifications[34], support the notion that the cellular milieu plays a pivotal role in shaping amyloid polymorphism. Once formed, tau polymorphs also display cell- and tissue-specific self-propagation and toxicity[11]. However, the extent to which tau amyloid polymorphism in end-stage disease is a reporter of early pre-amyloid pathology, a driver of tau spreading, toxicity, or all three, remains a matter of intense research. Understanding the thermodynamic underpinnings of tau structural polymorphism is therefore crucial to understand both the genesis, as well as the biological activity of tau amyloid polymorphs.

In this study, we employed a combination of aggregation experiments conducted on a comprehensive peptide collection spanning the complete tauRD amyloid core sequence, along with a computational analysis of known amyloid structures formed in tauopathy patient brains. This approach allowed us to map all segments of tauRD with high local amyloid propensity. Considering the non-uniform amyloidogenic nature of the tau RD sequence, a key question emerged regarding the respective contributions of amyloidogenic and non-amyloidogenic segments to the stability of tau amyloid structure formed from the intact tau sequence in pathogenic amyloid structures (Fig. 6a). We found that the segments with the highest amyloid propensity computed or determined here experimentally coincide with the thermodynamically most favourable elements in full-length tau amyloid cores and stabilise different polymorphs through key interactions. By delineating the impact of each of these segments, our study led to the identification of PAM4, a previously overlooked amyloid motif encompassing residues 350–362 of tau R4. Further characterisation of this region confirmed that it forms bona fide amyloid fibrils, and that it induces dose-responsive seeding of P301S tauRD in a tau

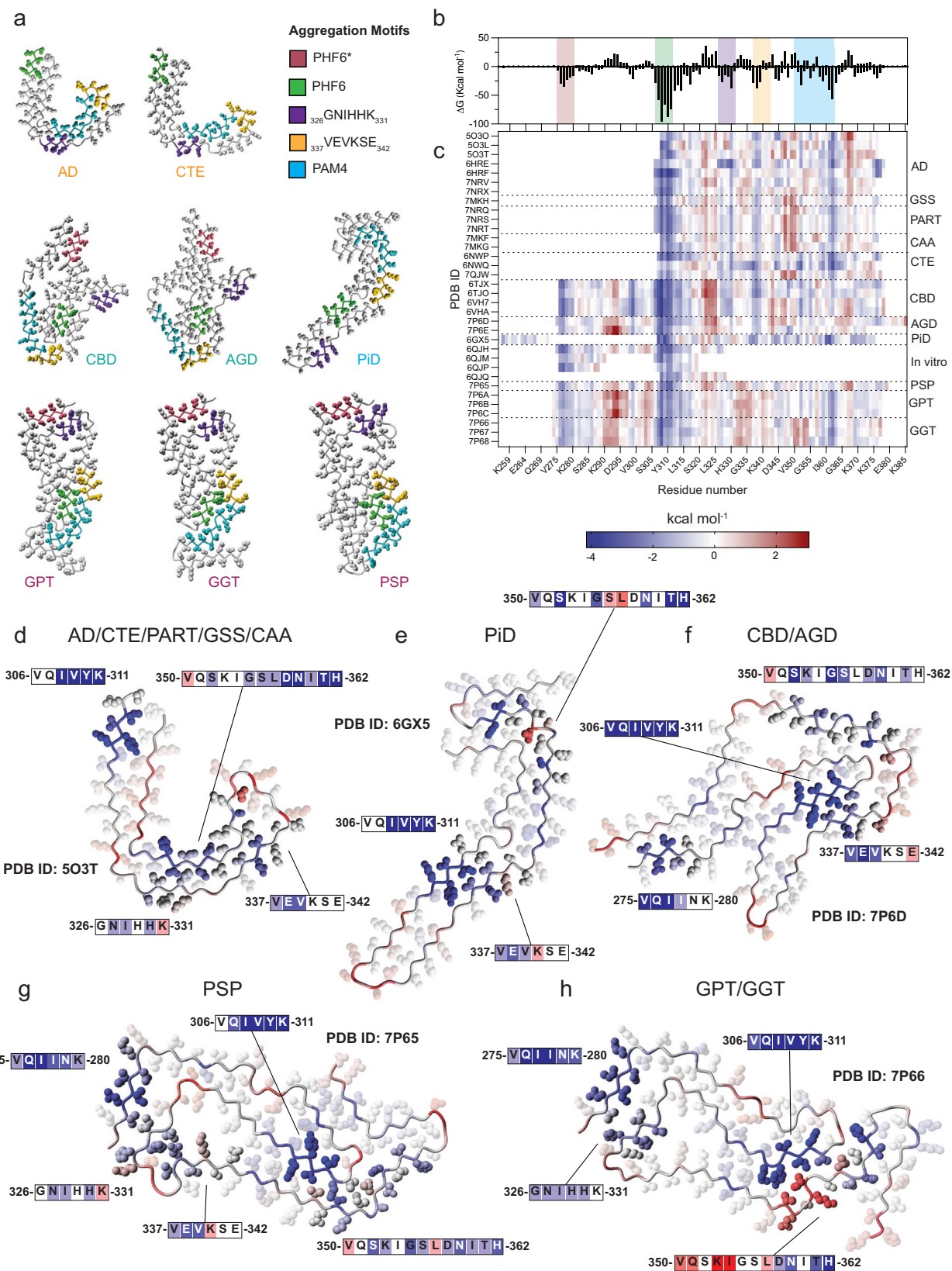

biosensor cell line. Subsequent structural determination of fibrils formed by the PAM4 peptide using cryo-EM revealed that they are intrinsically highly polymorphic, with the peptide adopting four different folds that closely match those seen in intact ex vivo tau fibril structures. The PAM4 fragment therefore forms discrete, but individually stable, amyloid folds that are not significantly altered in the context of the entire tauRD amyloid core, suggesting they are constructively integrated in tau polymorphs. This observation agrees with our thermodynamic analysis showing that PAM4 residues constitute stable yet conformationally distinctive structural elements in the different tau fibril polymorphs in which they are present (Figs. 5 and 6).

**Fig. 6 | Thermodynamic analysis of tau fibril polymorphs. a** Mapping the positions of identified aggregation motifs on tau fibrils polymorphs of major structural classes. **b**, **c** Heatmap plot indicating the per residue energy contributions for structurally determined tau amyloid fibril cores derived from different patient extracts or fibrils produced in vitro[45]. PDB IDs of individual tau fibril structures are shown on the Y-axis, separated by dashed lines based on disease type, whereas the X-axis indicates tau residue numbers. The top panel bar plot (**b**) indicates the sum of energy contributions per residue for all structures shown in the heatmap, with the identified aggregation motif regions shown in shaded boxes and coloured as in **a**. **d**–**h** Tau fibril polymorphs are stabilised primarily by the identified aggregation motifs. Residue sidechains are shown in ball representation, with important aggregation motif interactions highlighted and the rest of the residue sidechains faded out. Residues are coloured based on the calculated energies shown in **c**.

The significance of PAM4 in maintaining the stability of multiple polymorphic forms was further verified through seeding experiments involving a tauRD lacking PAM4 and tau seeds from patients with various tauopathies. Notably, the seeding efficiency of amyloid seeds purified from patients with AD, PSP, and to a lesser degree CBD, is notably diminished when incubated with HEK-293 cells expressing tauΔPAM4. Importantly, studies using molecular dynamic simulations further corroborate the role of PAM4 indicating that it is the primary region affecting the free energy of tau monomer dissociation from the ends of AD filaments[37]. In fact, experimental evidence on EGCG binding that disassembles tau filaments by introducing a wedging effect indicated a disappearance of density in the cryoEM maps corresponding to the peptide backbone at Gly355 in the middle of the PAM4 sequence, arguing that this could be an important starting step for tau fibril dissociation[38]. Together, this evidence is in agreement with our finding that PAM4 is a key amyloidogenic motif alternatively stabilising distinct pathological tau polymorphs.

The identification and characterisation of PAM4 also agrees with the output of different amyloid prediction algorithms including WALTZ[24], CORDAX[25], and zipperDB[30] suggesting that the [350]VQSKIG[355] and [358]DNITHV[363] hexapeptides have intrinsic amyloid propensity. However, these shorter peptides do not form amyloid fibrils unless they are joined in a unique segment encompassing residues 350–362; the full PAM4 motif. Inspection of the fibrillar PAM4 structures shows that the flexibility of the GS linker enables alternative backbone orientations perpendicular to the filament axis. It also reveals that internal PAM4 interactions, such as the K353-D358 salt bridge or interchain interactions stabilise the PAM4 polymorphs. In full tau amyloid cores, less stable residues of PAM4 are compensated by interfacing with other stabilising elements in tau folds, such as the PHF6 or [326]GNIHHK[331] aggregation-prone segments. Moreover, Gly355 in PAM4 introduces a ~70° turn that has been shown to be critical for establishing the β-helical geometry of AD fibrils[22]. This β-helix structure has been linked to promoting the prion-like propensity of C-shaped tau strains[22] and is further stabilised by the salt bridge between Lys353 and Asp358 stacked ladders[22,39] as observed in the FoldB conformer. Post-translational modifications may also mediate the influence of PAM4 on tau aggregation. For instance, phosphorylation of Ser356 has been shown to reduce in vitro fibrillation and cellular seeding of tau[40] suggesting that the structural flexibility provided by this segment is important for tau fibril stabilisation. Similarly, N-glycosylation of Asn359 has been shown to affect tau phosphorylation levels, whereas an N359Q mutation was suggested to reduce human tau accumulation in transgenic fly models[41]. Finally, PAM4 is also a potential target region for the development of molecular diagnostics and therapeutics, since it forms a binding pocket supporting small molecule binding with potential high affinity to tau fibrils, as shown in the case of PET ligands such as GTP-1[42] and MK-6240[43].

In conclusion, our study suggests that tau polymorphism is modular and geared by a common framework of amyloidogenic segments that is stabilised by alternative tertiary folds. While it is clear that physiological and pathological context shape tau polymorphism, we have identified PAM4 as an amyloidogenic segment which displays discrete intrinsic tertiary structural preferences restricting the polymorphic register of this segment in isolation as well as in the context of pathological tau amyloids. Our work provides rational insight into the thermodynamic make-up of tau polymorphs and shows how not only environmental cues but also intrinsic structural preferences of the tau sequences shape polymorphism in disease.

## Methods

### Computational predictions and analysis

Calculations of aggregation propensity were performed using state-of-art aggregation predictors, such as WALTZ[24] and Cordax[25]. WALTZ is based on a position-specific scoring matrix produced against a wide dataset of amyloidogenic sequences[44], whereas Cordax is a structure-based predictor that is less limited by typical propensities such as beta-propensity and hydrophobicity and as a result, is able to provide predictions that untether aggregation propensity from solubility.

Using a thermodynamic profiling method recently developed[20,45], we calculated the sum of all interactions made by each residue found in the amyloid core of tau fibril structures (main chain and side chain), both within the same polypeptide chain, as well as between different layers within a protofibril. These per residue energy calculations solely take protein interactions into account and do not factor in potential post-translational modifications or stabilising interactions from non-proteinaceous bound co-factors which could further influence the stability of the amyloid core structure. Consequently, the energy profiles generated report on the contribution of intrinsic polypeptide main-chain and side-chain interactions (i.e., cross-β quaternary structure and the way they are stabilised by tertiary interactions) to the stability of the amyloid core. Structural analysis, representation and alignments were performed using YASARA v21.12.19 and SuperPose[46].

### Preparation of peptide samples

The peptide library spanning the tau repeat domain residues was synthesised from GenScript, ensuring each peptide was of sufficient purity (>90%, Supplementary Figs. 9–S18) and with blocked ends (N-terminal acetylation, C-terminal amidation). Samples were initially pre-treated with 1,1,1,3,3,3-hexafluoro-isopropanol (HFIP) (Merck), then dissolved in 25 mM HEPES, 10 mM KCl, 5 mM MgCl$_2$, 3 mM TCEP, 0.01% NaN$_3$, pH 7.2 buffer, unless stated otherwise. A synthetic peptide corresponding to the PAM4 region was synthesised using an Intavis Multipep RSi solid phase peptide synthesis robot. Two peptide batches were prepared, one with peptide purity >99% and one with purity above 92%. For the latter, LCMS revealed the presence of an additional peak corresponding to an Fmoc-coupled PAM4 derivative (Supplementary Fig. 1). The highest purity peptide (>99%) was used for all experiments, except for cryo-EM where both batches were tested. Peptide aliquots were stored as ether precipitates (−20 °C). After initial HFIP pre-treatment, PAM4 samples were dissolved in 25 mM HEPES, 10 mM KCl, 5 mM MgCl$_2$, 3 mM TCEP, 0.01% NaN$_3$, pH 7.2 buffer, unless stated otherwise.

### Negative staining

Hits derived from the peptide screen library were incubated for 7 days at room temperature. A drop (5 μL) was applied to glow-discharged (50 s) 400-mesh carbon-coated copper grids for 60 s (Agar Scientific Ltd., England). The grids were subsequently washed several times with milli-Q water and negatively stained using a 2% (w/v) uranyl acetate solution (in milli-Q). A similar protocol was used for the preparation of PAM4 peptide grids, using a 5 mg/mL peptide solution dissolved in mQ

water and left to form mature fibrils for a week before application to 300-mesh continuous carbon grids (Agar Scientific Ltd., England). Grids were examined with a JEM-1400 120 kV transmission electron microscope (JEOL, Japan), operated at 80 keV.

### Thioflavin-T and pFTAA binding assays
Solutions of each peptide (200 µM) were prepared and analysed in half-area black 96-well microplates after a week of incubation to form mature fibrils (Corning, USA). For PAM4 kinetics, samples with various concentrations of the peptide were prepared in parallel, following filtering with 0.2 µM filters. Aggregation was assessed by adding Th-T (Sigma) or pFTAA (Ebba Biotech AB) at a final concentration of 25 µM and 0.75 µM, respectively. Fluorescence intensities were measured as independent repeats ($n = 3$) at 30 °C, with a double-orbital shaking step (100 rpm for 10 s) before each cycle, using a FLUOstar Omega plate reader (BMG Labtech, Germany), equipped with an excitation filter at 440 nm and corresponding emission filters at 490 and 510 nm, respectively.

### Congo red staining
Droplets (10 µL) of the PAM4 peptide solution containing mature amyloid fibrils were dried slowly in ambient conditions on glass slides in order to form thin films. The films were then stained with a Congo red (Sigma) solution (0.1% w/v) prepared in milli-Q water for 20 min. De-staining was performed with gradient ethanol solutions (70–90%). Imaging was performed on a SMZ800 stereomicroscope equipped with a polarising filter (Nikon) and a DS-2Mv digital camera (Nikon).

### X-ray diffraction
A single droplet of the PAM4 peptide solution (5 mg/ml dissolved in mQ) containing mature amyloid fibrils was dried between wax-covered capillary tubes to form oriented fibres suitable for X-ray fibre diffraction. X-ray diffraction patterns were collected with a Rigaku (Tokyo, Japan) copper rotating anode (RA-Micro7 HFM) operated at 40 kV and 30 mA, with a wavelength λ = 1.54 Å. The specimen-to-film distance was set to 200 mm and the exposure time was 1200 s. Diffraction patterns were studied using and displayed using the Adxv software (Scripps Research, USA).

### Fourier-Transform Infrared Spectroscopy (FTIR)
Suspensions (5 µL) of the PAM4 peptide (5 mg/ml dissolved in mQ) were cast onto a 96-well silicon microplate (Bruker) and air-dried to form thin films. Spectral scans (120) were acquired at 4 nm$^{-1}$ resolution in transmission mode and averaged to improve signal-to-noise ratio, using an HTS-XT FTIR microplate reader (Bruker). Background correction was performed by subtracting spectra obtained from a blank position of the microplate. Spectral normalisation and 2$^{nd}$ derivatives with a 13-point smoothing, using Savitzky-Golay filtering[47], were calculated using the OPUS software (v8.5.29).

### Atomic Force Microscopy (AFM)
Solutions containing fibrils formed by the PAM4 peptide (5 mg/ml dissolved in mQ) were diluted in milli-Q water to a final concentration of 1 mg/mL, then adsorbed to freshly cleaved mica for 15 min at room temperature. The mica was carefully washed 3 times (milli-Q water), to remove non-adsorbed material, and left to dry for 30 min at ambient conditions. For AFM scanning, RFESP-75 (with a normal frequency of 75 kHz, a spring constant of 3 N/m and tip radius of 8 nm) cantilever probes (Bruker) were used. Amyloid fibrils were imaged using a JPK NanoWizard Sense+ (Bruker) in oscillation mode (scan rate 0.5 Hz; amplitude setpoint 0.47 V), at a scan size of 10 × 10 um with an IGain of 300 Hz.

### Cryo-EM data collection
Fibrillated PAM4 peptide (5 mg/ml dissolved in mQ) was diluted to 2.5 mg/mL and Fmoc-PAM4 peptide batch (5 mg/ml dissolved in mQ)

to 0.6 mg/mL in milliQ water prior to application onto 60 s plasma cleaned (Tergeo, Pie Scientific) Lacey carbon 300 mesh grids. Both samples were blotted and frozen in liquid ethane using a Vitrobot Mark IV (FEI) with a 2 s wait and 5 s blot time respectively. The Vitrobot chamber was maintained at close to 100% humidity and 6 °C. Both cryo-EM datasets were collected at the University of Leeds Astbury centre using a Titan Krios electron microscope (Thermo Fisher) operated at 300 kV with a Falcon IV detector (Thermo Fisher) and Selectris energy filter set with a 10 e-V slit width (Thermo Fisher). A nominal magnification of 130,000× was set yielding a pixel size of 0.95 Å (acetyl-PAM4) and 0.94 Å (Fmoc-PAM4). For the acetyl-PAM4 dataset, a total of 1512 movies were collected with a nominal defocus range of −1.4 to −2.6 µm using a total dose of ~39 e$^-$/Å$^2$ over an exposure of 5 s, which corresponded to a dose rate of ~7 e$^-$/pixel/s. For the Fmoc-PAM4 dataset, a total of 1,957 movies were collected with a nominal defocus range of −1.2 to −2.4 µm using a total dose of ~32 e$^-$/Å$^2$ over an exposure of 5 s, which corresponded to a dose rate of ~5.7 e$^-$/pixel/s. Each movie was collected as EER frames and compressed and regrouped into Tif fractions, with 1539 frames regrouped into 36 fractions (~1.1 e$^-$/Å$^2$/fraction) for acetyl-PAM4 and 1204 fractions regrouped into 35 fractions (~0.9 e$^-$/Å$^2$/fraction) for Fmoc-PAM4 respectively.

### Cryo-EM data processing
Each movie stack was aligned, dose weighted and summed using motion correction in RELION4[48] and CTF parameters were estimated for each micrograph using CTFFIND v4.14[49]. Micrographs not containing fibrils were removed to give 1329 (high purity PAM4, Supplementary Fig. 4) and 1931 (Fmoc-PAM4, Supplementary Fig. 6) and micrographs for further processing. Fibrils from around fifty micrographs were manually picked in RELION and the extracted segments used to train automated filament segment picking in crYOLO[50]. Using an inter-box spacing of three helical repeats (~14 Å), a total of 442,856 (high purity PAM4) and 520,390 (Fmoc-PAM4) helical segments were extracted 3× binned in RELION (box dimensions of ~85 nm) for the first two rounds of 2D classification, after which the selected segments were re-extracted 2× binned (box dimensions of ~57 nm) for the third round of 2D classification. Throughout, the VDAM classification algorithm was used to separate out picking artefacts and unfeatured tubes to leave 325,180 (Fmoc-PAM4) and 286,927 (high purity PAM4) fibril segments for further processing. During the last round of 2× binned 2D classification, multiple morphologies were evident within each dataset (Supplementary Figs. 4 & S6), and the datasets were split into subset pools based on the apparent fibril morphology in the class averages. For the high purity peptide batch, 185,879, 81,035 and 28,909 segments were assigned to the 2PF, 3PF and 4PF subsets respectively (Supplementary Fig. 4c). For Fmoc-PAM4, 173,207 segments were grouped as simple twisting fibrils and 151,973 segments grouped as complex twisting fibrils (Supplementary Fig. 6c–e). All of the subsets were separately extracted unbinned with box dimensions of ~28 nm (300 pixels) before a final 2D classification run (Supplementary Fig. 4e & S6e) to remove obviously mis-assigned segments and picking artefacts (4120, 4455, 13,519 and 7116 segments removed from PAM4 2PF, PAM4 3PF, Fmoc-PAM4 "simple" and Fmoc-PAM4 "complex" subsets, respectively).

For starting 3D classification, initial 3D templates (Supplementary Figs. 4d & S6d) were generated from single 2xbinned 2D class averages of each polymorph using the relion_helix_inimodel2d command[51] along with measured helical crossover estimates from 3xbinned 2D class averages. For the Fmoc-PAM4 segment pool, the first "Simple" subset 3D classification run was initiated using initial model i) and the "Complex" subset 3D classification run with initial model iii). For both datasets, a series of initial 3D classifications were run to obtain better initial models and estimates of the helical twist for each subset before going back to the starting segment pools for the following described workflow. The

first 3D classification run on each subset with the updated initial models used a fixed twist (based on the updated estimate) and rise (4.80 Å), with a t-value of 20–30, 1.8° sampling, and a strict high-resolution limit of 5 Å with 2–4 output classes (Supplementary Figs. 5a, S5b, S7a & S7b). Details of the segments/classes selected for each morphology are shown in the processing schemes (Supplementary Figs. 5a–d & S7a–e). For the second round of 3D classification, helical searches of the twist were employed with a fixed rise (4.80 Å) with a t-value of 15–20, 1.8° sampling and a strict high-resolution limit of 4 Å with 2–3 output classes. In the third 3D classification run, narrow helical searches of both the twist and rise were employed with a t-value of 15–20, 0.9° sampling and a strict high-resolution limit of 3.5 Å with two output classes. The most ordered class was selected for each morphology from this final classification to move on to refinement.

Often, a couple of sequential rounds of 3D refinement were needed to get the halfmaps to converge before CTF refinement of the per-particle defocus estimates, Bayesian polishing and then final runs of 3D refinement. Typically, narrow helical searches of twist and rise were used up until the final refinement, with t-values of 10, initial sampling of 0.9° and initial lowpass-filtering of 6 Å. Post-processing with a soft mask (extended by 5 pixels and soft-edge for 8 pixels, z length of 20%) was used to obtain gold-standard resolution estimates at an FSC value of 0.143 for each final map. As such, the final PAM4 2PF (Type 1) map is deposited with a resolution of 2.6 Å and sharpening B-factor value of −35 Å$^2$. The PAM4 3PF and 4PF morphologies could not be resolved to high resolution, potentially due to increased heterogeneity in combination with fewer numbers of segments. The final Fmoc-PAM4 Type 2 map is deposited with a resolution of 2.8 Å and sharpening B-factor value of −39 Å$^2$, the Type 3 map with a resolution of 2.7 Å and sharpening B-factor of −50 Å$^2$ and the Type 4 map with a resolution of 2.9 Å and sharpening B-factor of −26 Å$^2$. The full collection and processing details are shown in Supplementary Table 2.

### Model building and refinement

A de novo model was built first for the unique peptide chains in one fibrillar layer of each of the PAM4 Type 1 and the PAM4 Type 2 cryoEM maps using Coot[52]. To avoid biasing the models, the multiple solved ex vivo Tau structures were purposefully not used to guide model building, and were only used for comparative purposes once the models had been built. Both Ramachandran and rotamer outliers were monitored and minimised during building in Coot. The PAM4 Type 2 model chains were used as a starting template for building of the Type 3 and 4 structures. For each structure, the final built asymmetric unit was then repeated and rigid body fit to generate a model for 3 layers of the fibril core and used for real space refinement against the deposited map in Phenix v1.17.1[53]. NCS restraints were applied to prevent divergence of repeating chains in the layers along the fibril axis, but the individual subunits within a layer were left to refine independently. The final real space refined models for each respective structure were assessed using MolProbity[54] and deposited, with the final model statistics summarised in Supplementary Table 2.

### Cryo-ET data collection and fibril hand determination

The resolution of the final cryoEM maps for each of the PAM4 structures (Types 1, 2, 3 and 4) was sufficient to unambiguously build left-handed fibrils based on the fit and orientation of the peptide backbone within the density (Supplementary Fig. 8a). As an additional check, tilt series of acetyl-PAM4 peptide fibrils were collected using a Titan Krios electron microscope (Thermo Fisher) operated at 300 kV with a Falcon IV detector (Thermo Fisher) and Selectris energy filter set with a 10 e$^-$V slit width (Thermo Fisher). The same grid was used as used for the respective cryoEM data collection in the previous sections. A nominal magnification of 64,000× was set yielding a pixel size of 1.93 Å. A total of 5 tilt series were collected using Tomo5.12 software (ThermoFisher) with a nominal defocus range of −4.5 to −5.0 μm using

a total dose of ~91 e−/Å$^2$ divided over 61 dose-symmetric tilt images from −60 to +60° with a step of 2°. Each tilt image was collected as a 6-frame movie with a dose of 0.25 e$^-$/Å$^2$ per frame and a total exposure of 1.3 s per tilt, corresponding to a dose rate of ~4.4 e$^-$/pixel/s. Tilt series were motion corrected and dose weighted using MotionCor2[48], stacked and reconstructed 3× binned (pixel size of 5.79 Å) into 3D tomograms using IMOD etomo[55] with fiducial-less alignment and a tilt axis rotation of 95.5°. An identical reconstruction protocol on the same microscope with a tilt angle of 95.5° was previously shown to faithfully determine the hand of amyloid fibrils, whereby the first z-slices of the resulting tomograms relate to the back/bottom face of the fibril and the later z-slices go up to the top/front face[56]. As such, comparing the early, middle and late z-slices of the tomograms of PAM4 Type 1 fibrils revealed a left-handed twist in all of the six fibrils checked. Three examples of the slices and twist assessment compared to slices through the high-resolution Type 1 fibril structure are displayed in Supplementary Figs. 8b and S8c. Movies of PAM4 tomograms along the z-axis (bottom-up, from back to the front of fibrils) are provided as Supplementary Movie S1 and S2.

### Cellular seeding assays

HEK-293 cells were cultured in DMEM medium, supplemented with 10% FBS, 1 mM sodium pyruvate and non-essential amino acids (Gibco), under an atmosphere of 5% CO$_2$ at 37 °C. Cells were plated at 15.000 cells/well (for 24 h seeding) or 10.000 cells/well (for 48 h seeding) in 96-well PhenoPlates (PerkinElmer) that were coated with poly-L-lysine (Sigma) for 30 min. Cells were transfected using lipofectamine 3000 according to the manufacturer's protocol. First, cells were transfected with 50 ng plasmid expressing tauRD-eYFP conjugates with a P301S mutation, w/o APR deletions. Following plasmid transfection (24 h), cells were transfected further with rTau (24 h incubation), peptide or patient derived extracted aggregates (48 h incubation). Just before transfection, samples were sonicated for 15 min (30 s on, 30 s off at 10 °C) with a Bioruptor Pico (Diagenode). Then, 5 μL of sample, mixed with 0.2 ul of 3000 reagent, was added to a mixture of 4.5 μL of Opti-MEM medium (Gibco) with 0.3 μL Lipofectamine 3000. After a 15 min incubation at room temperature, 10 μL of mixture was added per well. Cell medium was replaced with 100% ice-cold methanol and plates were incubated on ice for 15 min, then washed three times with PBS. Three individual plate preparations were performed per sample as independent experiments (n = 3). High-content screening was performed at the VIB Screening Core/C-BIOS, using an Opera Phenix HCS (PerkinElmer) equipped with proper filter channels to track tau aggregation through the YFP channel (Ex:490-515 nm, Em:525-580 nm). Image storage (16 fields in 3 planes at a 40× magnification were acquired per well) and segmentation analysis was performed using the Columbus Plus digital platform (PerkinElmer).

### Preparation of recombinant tau seeds

Recombinant full-length tau (tau$^{2N4R}$) was as described in previous protocols[57]. Lyophilised protein aliquots were freshly dissolved in 10 mM HEPES, pH 7.4, 100 mM NaCl at a final concentration of 10 μM. Following filtration, using 0.2 μM PVDF filters, 5 μM of heparin (Sigma) was added to the solution to induce aggregation. After 7 days of incubation at 37 °C under shaking (700 rpm), we generated seeds by breaking endpoint amyloid fibrils through successive sonication for 15 min (30 s on, 30 s off) at 10 °C, using a Bioruptor Pico sonication device (Diagenode).

### Extraction of tau aggregates

Ethical approval to access and work on the human tissue samples was given by the UZ Leuven ethical committee (Leuven/Belgium; File-No. S63759). An informed consent for autopsy and scientific use of autopsy tissue with clinical information was granted from all subjects involved

(Supplementary Table 1). Following approval, brain tissue from autopsy cases was received from UZ/KU Leuven Biobank and the University of British Columbia (UBC) (Supplementary Table 1). The tissue samples were all fresh-frozen samples of frontal cortex. Immunohistochemical evaluation of formalin-fixed, paraffin-embedded sections from the same anatomical region (i.e. frontal cortex) confirmed the presence of abundant tau-immunoreactive pathology with characteristic features of each condition. Sarkosyl-insoluble material was extracted from cortex tissue of four individual patients with Alzheimer's disease (AD1–AD4), as well as from individual patients with CBD, PSP, and PiD, as described in previous work[8,58]. Briefly, tissue homogenisation was performed with a FastPrep (MP Biomedicals) in 10 volumes (w/v) cold buffer (10 mM Tris-HCl pH 7.4, 0.8 M NaCl, 1 mM EGTA and 10% sucrose) and a centrifugation step at $20,000 \times g$ for 20 min at 4 °C. Universal Nuclease (Pierce) was added to the supernatant, followed by a 30 min incubation at room temperature. Subsequently, the sample was brought to 1% Sarkosyl (Sigma) and incubated for 1 h at room temperature while shaking (400 rpm), followed by centrifugation at $350,000 \times g$ for 1 h at 4 °C. The pellet was washed once, resuspended in 50 mM Tris-HCl pH 7.4 (175 mg of starting material per 100 μl) and stored at −80 °C.

### Reporting summary

Further information on research design is available in the Nature Portfolio Reporting Summary linked to this article.

## Data availability

The PAM4 amyloid fibril cryoEM maps and models for the four solved structures are deposited in the EMDB and PDB respectively with codes: EMD-16876, PDB-8OH2 (PAM4 Type 1); EMD-16881, PDB-8OHI (PAM4 Type 2); EMD-16883, PDB-8OHP (PAM4 Type 3); EMD-16886, PDB-8OI0 (PAM4 Type 4). The following PDB files were used in this study: 5O3O, 5O3L, 5O3T, 6HRE, 6HRF, 7NRV, 7NRX, 7MKH, 7NRQ, 7NRS, 7NRT, 7MKF, 7MKG, 6NWP, 6NWQ, 7QJW, 6TJX, 6TJO, 6VH7, 6VHA, 7P6D, 7P6E, 6GX5, 6QJH, 6QJM, 6QJP, 6QJQ, 7P65, 7P6A, 7P6B, 7P6C, 7P66, 7P67, 7P68. Source data are provided with this paper.

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

## Acknowledgements

The authors would like to thank the VIB BioImaging Core for training, support and access to the instrument park, the VIB Screening Core/C-BIOS facility for excellent support with cellular screening and the Laboratory for Biocrystallography (KU Leuven) for support with X-ray diffraction experiments. This work was supported by the Flanders institute for Biotechnology (VIB); KU Leuven (grant C3/20/057); the Fund for Scientific Research Flanders (FWO; project grants G0C3522N, G053420N and G045920N, FWO/Hercules Foundation equipment grants I011220N and NextGenQBio - AH2016.133, PhD fellowship 1163823N to G.T. and Postdoctoral Fellowships 12P0919N and 12P0922N to N.L.); the Stichting Alzheimer Onderzoek (SAO-FRA 2020/0030, SAO-FRA 2020/0013, and SAO-FRA 2019/0015). Research reported in this publication was also supported by the National Institute On Aging of the National Institutes of Health under Award Number R01AG079234. The content is solely the responsibility of the authors and does not necessarily represent the official views of the National Institutes of Health. Neuropathological characterisation of human brain samples was supported by FWO grants (G0F8516N and G065721N) to D.R.T. M.W., S.E.R. & N.A.R. thank the UK Medical Research Council (MR/T011149/1) for support. SER holds a Royal Society Professorial Fellowship (RSRP \R1\211057). CryoEM was performed at the Astbury Biostructure Laboratory, which was funded by the University of Leeds and Wellcome (221524/Z/20/Z). We thank Rebecca Thompson, Emma Hesketh, Yehuda Halfon, Louie Aspinall and Josh White for cryoEM support. All data processing was performed using the ARC4 compute cluster at the University of Leeds. Finally, we thank colleagues in the Radford and Ranson laboratories for many helpful discussions while preparing this manuscript. For the purpose of Open Access, the authors have applied a CC BY public copyright license to any Author Accepted Manuscript version arising from this submission.

## Author contributions

N.L, F.R, and J.S. conceived and planned the experiments. N.L., M.W., G.T., M.R., C.M., S.D'H., and V.G. performed the experiments. N.L., M.W., performed data analysis. N.L. performed computational analysis. N.L., M.R., and T.G. contributed to sample preparation. N.L., R.G., D.A., N.R., S.R., F.R., and J.S. contributed to the interpretation of the results. D.R.T., I.R.M., and R.R. provided patient samples. N.L., F.R., and J.S. wrote the manuscript and all authors provided critical feedback, helped shape the research, and contributed to the final version.

## Competing interests

The authors declare no competing interests.
