## [Peer Review File · Nature Communications]

REVIEWER COMMENTS

Reviewer #1 (Remarks to the Author):

This manuscript has improved after my original review and now includes additional controls and cryo-EM data. I support its publication, and only have minor comments below.

L304-306: "In conjunction to our cell assays showing that PAM reduced AD, PSP and CBD seeding, our findings here show that the cryo-EM structures of PAM4 amyloid polymorphs in isolation can be leveraged to understand the structure-activity relationship between amyloid core structure and its amyloid templating in cells"

=> This is an unnecessary stretch that remains to be proven in the general case. I would therefore recommend to rephrase this sentence to just point out the agreement between the 2 types of experiments.

L328: Tau polymorphism arises from amyloid motif rearrangement into tertiary folds?

=> This has a question mark, but is grammatically a statement. Also, probably because it is a question, this section feels more like a Discussion than a Results section.

In the new Figure 5, some atoms are red others aren't. I would also suggest to make a zoomed-in view of the PAM4 overlaid on the disease-specific structures, as currently it is hard to see the differences. Perhaps only using Calpha-traces would help? Removing the ball-n-stick representation to a stick-only representation in the current Fig 5 would also help.

PS: In science there is no solace in numbers and the observation that many others do something does not in itself make it the right thing to do... ;-)

Reviewer #2 (Remarks to the Author):

The authors have addressed all of my concerns in their extensive revisions.

Reviewer #3 (Remarks to the Author):

In the manuscript from Louros and associates, the authors identify a new amyloidogenic motif in tau they refer to as Polymorphic Amyloid Motif of Repeat 4 (PAM4). Using a combination of approaches to characterize its role in tau assembly, the authors provide compelling data showing that the PAM4 peptide assembles into filaments that resemble disease-associated tau structures. The results are very intriguing and will significantly impact the field. However, there are a couple of issues that should be addressed prior to publication.

1. The authors compare the seeding activity of different tau peptides in Figure 2i-j, but as a high purity peptide containing the PAM4 region was synthesized for characterization, it would be important to ensure all peptides included in the comparison were synthesized to a similar purity. In addition, as seeding with the PAM4 peptide was assessed following sonication of mature PAM4 peptide filaments, it is important to ensure all peptides were treated equally prior to exposure to tauRD-transfected HEK293 cells. Moreover, as the peptide 348-362 was characterized in Figure 1, while the peptide 350-362 (PAM4) was generated and examined in Figure 2, it would seem to be important to assess how the peptides 348-362 and 350-362 (PAM4) compare in the assays performed in Figure 1. In the seeding activity assay, the PAM4 peptide seems to be more effective than 348-362, but it is not clear whether this may be due to differences in purity or sonication of filaments prior to seeding.

2. The authors use cryo-EM to demonstrate that the high purity PAM4 peptide predominantly forms Type 1 filaments, while the lower purity PAM4 peptide batch forms Type 2-4 filaments. However, as Type 2-4 filaments are more representative of disease-associated tau polymorphs, it is unclear why the lower purity peptide was not used for other assays. In addition, can the authors speculate why the high purity PAM4 peptide fails to form the type of filaments observed in the brain, while the lower purity peptide is able to do this?

Minor comments

3. Please clarify the significance of underlined sequences in Figure 1e in the figure legend; presumably these are referring to the sequences identified in Figure 1b.

Response to reviewers

Reviewer #1 (Remarks to the Author):

This manuscript has improved after my original review and now includes additional controls and cryo-EM data. I support its publication, and only have minor comments below.

Response: We thank the reviewer for their constructive criticism that has helped to improve our manuscript.

L304-306: "In conjunction to our cell assays showing that PAM reduced AD, PSP and CBD seeding, our findings here show that the cryo-EM structures of PAM4 amyloid polymorphs in isolation can be leveraged to understand the structure-activity relationship between amyloid core structure and its amyloid templating in cells"

=> This is an unnecessary stretch that remains to be proven in the general case. I would therefore recommend to rephrase this sentence to just point out the agreement between the 2 types of experiments.

Response: We have rephrased this sentence according to the reviewer's suggestion (lines 310-311).

L328: Tau polymorphism arises from amyloid motif rearrangement into tertiary folds?

=> This has a question mark, but is grammatically a statement. Also, probably because it is a question, this section feels more like a Discussion than a Results section.

Response: We thank the reviewer for this comment. We have removed the question mark from the subtitle, but have kept this segment in the results section, as it also presents data derived from thermodynamic calculations of tau protein cores that were generated using a proprietary method we have previously developed.

In the new Figure 5, some atoms are red others aren't. I would also suggest to make a zoomed-in view of the PAM4 overlaid on the disease-specific structures, as currently it is hard to see the differences. Perhaps only using Calpha-traces would help? Removing the ball-n-stick representation to a stick-only representation in the current Fig 5 would also help.

Response: All of the proposed changes have been incorporated in Fig. 5. We thank the reviewer for their suggestions.

Reviewer #2 (Remarks to the Author):

The authors have addressed all of my concerns in their extensive revisions.

Response: We are grateful to the reviewer for their detailed and constructive comments.

Reviewer #3 (Remarks to the Author):

In the manuscript from Louros and associates, the authors identify a new amyloidogenic motif in tau they refer to as Polymorphic Amyloid Motif of Repeat 4 (PAM4). Using a combination of approaches to characterize its role in tau assembly, the authors provide compelling data showing that the PAM4 peptide assembles into filaments that resemble disease-associated tau structures. The results are very intriguing and will significantly impact the field. However, there are a couple of issues that should be addressed prior to publication.

Response: We thank the reviewer for the valuable comments provided. Below, we have addressed the concerns raised by the reviewer hoping to improve the clarity of our findings and the manuscript.

Point 1. The authors compare the seeding activity of different tau peptides in Figure 2i-j, but as a high purity peptide containing the PAM4 region was synthesized for characterization, it would be important to ensure all peptides included in the comparison were synthesized to a similar purity. In addition, as seeding with the PAM4 peptide was assessed following sonication of mature PAM4 peptide filaments, it is important to ensure all peptides were treated equally prior to exposure to tauRD-transfected HEK293 cells. Moreover, as the peptide 348-362 was characterized in Figure 1, while the peptide 350-362 (PAM4) was generated and examined in Figure 2, it would seem to be important to assess how the peptides 348-362 and 350-362 (PAM4) compare in the assays performed in Figure 1. In the seeding activity assay, the PAM4 peptide seems to be more effective than 348-362, but it is not clear whether this may be due to differences in purity or sonication of filaments prior to seeding.

Response to Point 1: Indeed, it is important that the above are clear to the readers. All peptide samples were prepared and treated using the same protocols prior to transfection. Briefly, as stated in the Methods section, all peptides were dissolved in parallel in the same buffer preparation (25mM HEPES, 10mM KCl, 5mM MgCl₂, 3mM TCEP, 0.01% NaN₃, pH 7.2) and incubated for 7 days at room temperature to form mature amyloid fibrils. The mature samples were then sonicated using a matching protocol in a Bioruptor Pico (15 min - 30 sec on, 30 sec off at 10°C) and then transfected using the same transfection protocol. Therefore, the observed differences are not expected to have occurred due to treatment, as all samples were prepared in parallel under the same conditions.

Individual HPLC chromatograms and the corresponding MS analysis performed during peptide quality control are now provided as additional supplementary material indicating the purity and masses detected for every peptide synthesized and used in our work (Suppl. Figures 9-18). As seen in the provided chromatograms, all peptides were synthesized to high purities (>90%). In fact, most of the peptide hits shown in Fig. 2i-j have comparable purity levels (>94-99%) to the PAM4 peptide. Importantly, as the differences observed in seeding efficiencies were 10- to 100-fold lower compared to PAM4, it is very unlikely that such a large difference in effect is due to the small differences in purity.

During screening in Fig. 1, we showed that the 348-362 fragment forms fibrils that bind amyloid reporter dyes and have typical amyloid-like characteristics, as observed by TEM. We have now added end-state values for the kinetics shown in Fig. 2A, so both dye binding

properties and the amyloid characteristics observed by TEM of fibrils formed by PAM4 are reported in Figure 2 (Fig. 2A and F). Considering the reviewer's comment, we have also incorporated a brief comparison statement for both the Th-T and TEM observations of the two peptides in the main manuscript (Pages 6-7, lines 160-163 and 172-174).

Point 2. The authors use cryo-EM to demonstrate that the high purity PAM4 peptide predominantly forms Type 1 filaments, while the lower purity PAM4 peptide batch forms Type 2-4 filaments. However, as Type 2-4 filaments are more representative of disease-associated tau polymorphs, it is unclear why the lower purity peptide was not used for other assays. In addition, can the authors speculate why the high purity PAM4 peptide fails to form the type of filaments observed in the brain, while the lower purity peptide is able to do this?

Response to Point 2: Starting from the latter, it is very important to point out that the higher purity PAM4 batch did not fail to produce representative folds to the ones observed in the brain. In fact, it formed filaments in which PAM4 adopts the FoldB conformation, which is representative of PiD and PSP tau disease polymorphs. The aggregation assays shown in Fig. 2 were performed simply to establish the amyloidogenic properties of the PAM4 segment in isolation as an identified APR of tau. As such, to ensure that the low percentage of impurity does not affect the overall aggregation propensity of the PAM4 region, we tested the high-purity peptide, which definitively illustrates the amyloidogenic properties of the PAM4 region.

Recent work suggests that compared to an N-terminally acetylated end, an Fmoc protection group can provide additional π - π stacking interactions (Sharma R. et al. 2022 ChemBioChem 23, e202200499). Interestingly, similar interactions seem to mediate protofilament interfaces in the case of type 2-4 filaments, which could potentially explain the additional filaments recovered from the lower purity peptide batch. Prompted by the reviewer's suggestion, a brief sentence speculating on the above has been added to the main manuscript (Page 10, lines 272-277).

Point 3. Please clarify the significance of underlined sequences in Figure 1e in the figure legend; presumably these are referring to the sequences identified in Figure 1b.

Response to Point 3: We are grateful to the reviewer for pointing this out. As correctly spotted, the underlined segments represent the sequences identified in Figure 1b. We have added this information in the corresponding figure legends (lines 917 and 937-938).